# Genotype and Phenotype Characterization of *Rhinolophus* sp. Sarbecoviruses from Vietnam: Implications for Coronavirus Emergence

**DOI:** 10.3390/v15091897

**Published:** 2023-09-08

**Authors:** Sarah Temmam, Tran Cong Tu, Béatrice Regnault, Massimiliano Bonomi, Delphine Chrétien, Léa Vendramini, Tran Nhu Duong, Tran Vu Phong, Nguyen Thi Yen, Hoang Ngoc Anh, Tran Hai Son, Pham Tuan Anh, Faustine Amara, Thomas Bigot, Sandie Munier, Vu Dinh Thong, Sylvie van der Werf, Vu Sinh Nam, Marc Eloit

**Affiliations:** 1Pathogen Discovery Laboratory, Institut Pasteur, Université Paris Cité, 75015 Paris, France; 2Institut Pasteur, The OIE Collaborating Center for the Detection and Identification in Humans of Emerging Animal Pathogens, Université Paris Cité, 75015 Paris, France; 3National Institute of Hygiene and Epidemiology, Hanoi 100000, Vietnam; 4Structural Bioinformatics Unit, Institut Pasteur, CNRS UMR3528, Université Paris Cité, 75015 Paris, France; 5Institut Pasteur, G5 Evolutionary Genomics of RNA Viruses, Université Paris Cité, 75015 Paris, France; 6Institut Pasteur, Bioinformatics and Biostatistics Hub, Université Paris Cité, 75015 Paris, France; 7Institute of Ecology and Biological Resources, Vietnam Academy of Science and Technology (VAST), Hanoi 70072, Vietnam; 8Molecular Genetics of RNA Viruses Unit, Institut Pasteur, CNRS UMR 3569, Université Paris Cité, 75015 Paris, France; 9Institut Pasteur, National Reference Center for Respiratory Viruses, Université Paris Cité, 75015 Paris, France; 10Ecole Nationale Vétérinaire d’Alfort, University of Paris-Est, 77420 Maisons-Alfort, France

**Keywords:** *Sarbecovirus*, horseshoe bat, virus evolution, spillover

## Abstract

Bats are a major reservoir of zoonotic viruses, including coronaviruses. Since the emergence of SARS-CoV in 2002/2003 in Asia, important efforts have been made to describe the diversity of *Coronaviridae* circulating in bats worldwide, leading to the discovery of the precursors of epidemic and pandemic sarbecoviruses in horseshoe bats. We investigated the viral communities infecting horseshoe bats living in Northern Vietnam, and report here the first identification of sarbecoviruses in *Rhinolophus thomasi* and *Rhinolophus siamensis* bats. Phylogenetic characterization of seven strains of Vietnamese sarbecoviruses identified at least three clusters of viruses. Recombination and cross-species transmission between bats seemed to constitute major drivers of virus evolution. Vietnamese sarbecoviruses were mainly enteric, therefore constituting a risk of spillover for guano collectors or people visiting caves. To evaluate the zoonotic potential of these viruses, we analyzed in silico and in vitro the ability of their RBDs to bind to mammalian ACE2s and concluded that these viruses are likely restricted to their bat hosts. The workflow applied here to characterize the spillover potential of novel sarbecoviruses is of major interest for each time a new virus is discovered, in order to concentrate surveillance efforts on high-risk interfaces.

## 1. Introduction

Bats constitute a reservoir of many viruses, among which several have the potential to cross the species barrier from their natural hosts and to infect humans and domestic animals, directly or via an intermediate host [1]. In particular, bats are a reservoir of three of the ten virus groups that present pandemic potential, as designated by the World Health Organization: henipaviruses, filoviruses and coronaviruses [2]. Indeed, during recent decades, several coronaviruses emerged from their bat hosts and were responsible for multiple and severe human and animal epidemics or pandemics. These include severe acute respiratory syndrome (SARS) [3], Middle East respiratory syndrome (MERS) [4] and the coronavirus disease (COVID-19) in humans; and swine acute diarrhea syndrome (SADS) [5] and porcine epidemic diarrhea virus (PEDV) in pigs [6]. More than 4000 coronaviruses from 14 different bat families have been identified so far [7], a number that is probably underestimated given the richness of bat species distributed worldwide and the recurrent discovery of novel bat-associated coronaviruses [8,9,10,11,12,13,14]. Since their distribution and spillover potential are usually unknown, some of these viruses may present public and animal health concerns, and a global health approach is needed to better evaluate the risk of emergence of such coronaviruses [15].

In particular, the recent emergence of SARS-CoV-2, which led to the COVID-19 pandemic, highlighted the importance of monitoring the circulation of bat-associated sarbecoviruses worldwide. In fact, serological evidence of prior exposure of humans to bat sarbecoviruses in rural China suggested that wildlife-to-human spillovers from bats may occur frequently but silently [16,17,18]. This evidence relies mainly on antigen/antibody binding methods with questionable specificity, especially when the viral nucleoprotein is targeted. By contrast, very specific methods such as seroneutralization did not reveal significant cross-species transmissions of bat sarbecoviruses to humans [19]. However, we cannot exclude that such events sometimes occur and could represent opportunities for virus adaptation that may lead to better human-to-human transmission. The necessity of increasing surveillance of bat sarbecoviruses has therefore emerged. Most novel bat-related sarbecoviruses have been associated with horseshoe bats from Asia, and to a lesser extent from Europe [20,21,22] and Africa [23,24]. For example, surveys of bats in China have revealed high diversity and prevalence (5–10%) of sarbecoviruses in different rhinolophid bat species [25,26]. Bat precursors of SARS-CoV-2 have been identified in *Rhinolophus* sp. bats from Laos [27], Thailand [28], Cambodia [29], South Korea [30] and Japan [31]. For the majority of these novel viruses, it is still unknown if they are able to infect various hosts (including humans) apart from their bat reservoir.

Sarbecoviruses use the cellular angiotensin-converting enzyme 2 (ACE2) receptor to enter and infect mammalian cells [32,33]. The spike viral sequence, and especially the receptor-binding domain (RBD), are major determinants of host range and tropism [34]. ACE2 is a ubiquitous receptor present in all vertebrate animals, and several studies demonstrated that it is an efficient *Sarbecovirus* receptor in both bat reservoirs and susceptible mammalian hosts [35], such as palm civets [36], pangolins [37] and mustelids [38]. Several bat-associated sarbecoviruses were shown to use the ACE2 receptor to enter mammalian cells [39,40,41], while most of them presented deletions in their RBDs, precluding their use of ACE2 as a cellular receptor. However, some bat-associated sarbecoviruses were shown to infect human and bat cells through an ACE2-independent entry [42]. Recently, we demonstrated that the bat-associated *Rh. marshalli* BANAL-236 sarbecovirus, phylogenetically close to SARS-CoV-2, was able to use ACE2 to infect mammalian cells but presented an enteric tropism in primates contrary to its SARS-CoV-2 relative, possibly due to its lack of a furin cleavage site [19,27]. Therefore, several factors other than the ability to bind to ACE2 may influence host tropism and the restriction of certain sarbecoviruses to specific host species [43,44,45].

In Vietnam, the prevalence of coronaviruses circulating among bats is high and was estimated to be between 22% [46] and 75% [47] depending on the species and location of sampling. In a previous study conducted in Southern Vietnam, thirty-four *Pedacovirus* sequences close to Scotophilus bat CoV-512 were identified in *Scotophilus kuhlii* bats roosting in bat guano farms [48]. A more recent study, also implemented around bat guano collection sites and pig farming sites, identified progenitors of pig-infecting *Pedacovirus* in bats roosting at the human–wildlife–domestic animal (i.e., pig) interface [49]. Several *Merbecovirus-*, *Tegacovirus-* and *Nobecovirus*-related viruses were also identified, but no *Sarbecovirus*. Here, we report the first identification and characterization of *Rhinolophus* sp. sarbecoviruses from Northern Vietnam. We performed molecular and genetic characterization of these viruses through broad-range phylogenetic analyses, and evaluated their phenotype regarding their spillover potential to humans and wild and domestic animals by testing in silico and in vitro the ability of their RBDs to bind to mammalian ACE2s, a major determinant of host tropism. We conclude that these viruses are likely restricted to their bat hosts and are prone to recombinations and interspecies transmissions than can favor the emergence of viruses with an extended host spectrum or tropism.

## 2. Materials and Methods

### 2.1. Ethical and Regulatory Statements

This study, including bat sampling and handling, was approved by the Ministry of Health (Vietnam) on the 21 September 2020 (reference 4026/QD-BYT). The detailed protocol was submitted to the Institutional Review Board (IRB) of the National Institute of Hygiene and Epidemiology (NIHE), which stated that the study was exempted from IRB request, on the 10 September 2020. All animals were captured, handled, sampled and released following previously published protocols and ASM guidelines [50]. All participants were protected by personal protective equipment that comprised protective clothing with long sleeves and covering the head, neck, arms, shoulders and upper body parts. Exportation from Vietnam and importation to France were conducted according to national regulations.

### 2.2. Bat Capture and Sample Collection

Two bat surveys were conducted from 22 February to 3 March and from 4 July to 13 July 2021 in 9 caves from two districts of Sơn La Province, Vietnam: Sốp Cộp from 22 to 26 February and Mộc Châu from 27 February to 3 March and from 9 to 13 July; and Sơn La City from 4 to 8 July (Figure 1, Appendix A).

Bats were morphologically identified as follows: photographs and selected external measurements of every captured bat were taken for identification based on morphological diagnoses in comparison with published descriptions and confirmed features of *Rhinolophus thomasi* from Cuc Phuong National Park, Northern Vietnam. Morphologically, *Rhinolophus thomasi* is distinguished from other horseshoe bat species in Vietnam and surrounding countries by a complex of external characteristics including a short lancet with a rudimentary tip; a rounded connecting process; horseshoe does not cover the muzzle completely and has well-developed lateral leaflets; and a lower lip that has three mental grooves [51,52].

To confirm the morphological identifications, we took advantage of the concomitant sequencing of bat transcriptome and used the Barcode of Life Data Systems (BOLD) as previously described [53]. Briefly, all trimmed reads were mapped onto the Cytochrome Oxydase subunit I (COI) database, followed by de novo assembly of mapped reads. The resulting contigs were uploaded to the BOLD Identification System (https://www.boldsystems.org/index.php/IDS_OpenIdEngine) (accessed on 7 August 2023) for species identification. When a discrepancy between BOLD-based and morphology-based species identification was observed, we determined the most probable species by positioning the COI sequence of the bat individual into a phylogenetic tree based on 103 sequences of the N-terminal region of the Cytochrome Oxidase I gene (region of 657 bp), representative of 35 *Rhinolophus* species.

### 2.3. RNA Extraction and Pan-Coronavirus PCR Screening

Total RNA was extracted from 100 µL of the samples using the NucleoSpin^®^ 8 Virus kit (Macherey-Nagel, Düren, North Rhine–Westphalia, Germany) according to the manufacturer’s instructions. Extracted RNA was stored at −80 °C until further analysis. Reverse transcription of RNA extracts was performed with the SuperScript IV RT kit (Invitrogen, Carlsbad, CA, USA) using random hexamers and according to the manufacturer’s recommendations. Samples were screened for the presence of coronaviruses with the pan-coronavirus nested PCR system targeting the viral polymerase, as previously described [54], and confirmed via Sanger sequencing.

### 2.4. Whole Genome Sequencing of Bat Sarbecoviruses

Complete *Sarbecovirus* genomes were obtained via *Betacoronavirus*-enriched next-generation sequencing, as previously described [27]. Briefly, specific *Betacoronavirus* primers were mixed with random hexamers and used in a reverse transcription reaction, followed by a MALBAC random amplification. NGS libraries were prepared for each individual positive bat sample using the DNA Prep (M) Tagmentation^®^ kit (Illumina), and sequenced onto NextSeq500 or NextSeq2000 Illumina sequencers, at an average depth of 80 million 150 bp or 100 bp single reads per sample. When needed, conventional PCR and Sanger sequencing were used to fill the gaps in the genomes (primers are available on request).

### 2.5. Molecular and Phylogenetic Analyses

A comprehensive database of complete *Sarbecovirus* genomes was first built based on two recent studies [23,55] and completed with other sequences manually picked from GenBank and GISAID (Appendix A) to cover non-human (bat, pangolin, palm civet and ferret badger) sarbecovirus diversity. A total of 221 complete genomes were selected. Several human prototype strains of SARS-CoV and SARS-CoV-2 were also included in the final dataset. For each complete genome, 5′-3′ UTRs and intergenic regions were manually removed and genes were concatenated before being translated with C2A.A2C [56]. Protein sequences were then aligned with MAFFT (in auto mode, global alignment [57]) and backtranslated in nucleotides using the original sequences. Given the high variability in *Sarbecovirus* sequences, this method ensured the correct alignment of selected whole genomes. The same strategy was applied to align Vietnamese *Sarbecovirus* genomes.

Phylogenetic analyses were performed at the gene level, either for Vietnamese isolates or for the complete *Sarbecovirus* dataset, with W-IQ-TREE [58], using the maximum likelihood reconstruction method and the aBayes branch support parameter, after determination of the best substitution model fitting the data implemented in W-IQ-TREE. Trees were visualized with FigTree [59].

Identity matrices (expressed in %) were calculated at the whole genome and at each gene nucleotide and amino-acid level using CLC Main Workbench 21.0.4 (Qiagen, Hong Kong, China).

Recombination analyses were conducted on the Vietnamese or on the complete *Sarbecovirus* datasets using the “full exploratory recombination scan” implemented in the Recombination Detection Program (RDP v.5.36) [60], which includes RDP [61], GENECONV [62], MaxChi [63], Chimaera [64] and 3Seq [65] tools. Only recombination events detected with at least three of the above cited methods were considered true recombination events.

### 2.6. In Silico Evaluation of RBD-ACE2 Binding Free Energy

Bat ACE2 sequences missing from the databases at the time of analysis (i.e., *Rh. malayanus*, *Rh. thomasi* and *Chaerephon plicatus*) were reconstructed from our whole host genome sequences and completed with public data collected from NCBI/SRA. Briefly, a custom database comprising ACE2 sequences from Chiroptera or *Rhinolophus* sp. was created and queried using NCBI/SRA and our data using Diamond BlastX to identify ACE2-associated sequences. The assembly of identified sequences was then performed with Megahit.

The ACE2 genes from *Rhizomys pruinosus* (hoary bamboo rat) and *Hystrix brachyura* (Malayan porcupine) were similarly retrieved from genome assembly (respectively GenBank GCA_009823505.1 and GCA_016801275.1) with BlastN against a custom database of Rodentia ACE2 genes.

To estimate the affinity between each of the *Sarbecovirus* RBD and ACE2 receptors studied here, we used the following in silico pipeline:We used AlphaFold2 [66] to build structural models of ACE2 and RBDs. For each sequence, we generated 5 different models and selected the best model based on the AlphaFold2 pLDDT score.For each potential RBD-ACE2 complex, we created 200 models of the complex using HADDOCK [67]. ACE2 and RBDs were first docked into a complex with the help of inter-subunit distance restraints. These restraints were defined between each pair of atoms at the interface of the SARS-CoV-2 RBD and human ACE2 based on the X-ray structure of this complex (PDB code 6M0J [68]). The generated models were then refined via molecular dynamics simulations in explicit solvent during the last step of the HADDOCK modeling pipeline.The FoldX v. 5 scoring function [69] was used to estimate the RBD-ACE2 binding free energy from the ensemble of models generated with HADDOCK.

### 2.7. Generation of Lentiviral Pseudoviruses and ACE2-Dependent Entry Assays

Human codon-optimized sequences corresponding to the spike ectodomains of RtVN21-29, RtVN21-192, RtVN21-193 and RtVN21-201 were synthetized (GenScript) and cloned into a pVAX1 vector using a 2-fragment In-Fusion reaction with a sequence corresponding to the transmembrane and the C-terminal domains, with the 19 last amino acids deleted (positions 1222-1240 for RtVN21-29, positions 1224-1242 for RtVN21-192, positions 1216-1234 for RtVN21-193 and positions 1223-1241 for RtVN21-201).

Spike-pseudotyped lentiviral particles were produced via co-transfection in HEK-293T cells of the spike expression plasmid (5 µg of pVAX1-S-RtVN21-29, -192, -193, -201) together with a lentiviral backbone expressing the firefly luciferase (10 µg of pHAGE-CMV-Luc2-IRES-ZsGreen-W) and lentiviral helper plasmids encoding Gag-Pol, Tat and Rev (3.3 µg each of HDM-Hgpm2, HDM-tat1b and RC-CMV-Rev1b) using calcium phosphate precipitation [70]. Lentiviral particles were recovered 48 h post-transfection, clarified and tested for entry in HEK-293T expressing the human ACE2 receptor. Briefly, spike-pseudotyped lentiviral particles were serially diluted 3-fold and 50 µL was mixed with 50 µL of a HEK-293T-hACE2 cell suspension containing 4.10^5^ cells/mL and incubated in a 96-well white plate for 72 h. Firefly luciferase activity was then assessed by adding 100 µL of BrightGlo substrate (Promega, Madison, WI, USA), and luminescence was measured using a Berthold Centro XS luminometer.

## 3. Results

As part of a global study assessing the viral diversity carried by horseshoe bats from Northern Vietnam, we report in this study the identification and the molecular characterization of seven novel sarbecoviruses.

### 3.1. Sarbecovirus Detection Rate in Vietnamese Bats

A total of 214 insectivorous bat rectal swabs were collected in Sơn La Province between February and July 2021 (22 February to 3 March and 4–13 July). They belonged to 19 bat species: *Aseliscus stoliczkanus* (N = 15), *Barbastrella* cf. *darjelingensis* (N = 1), *Hipposideros armiger* (N = 47), *Hipposideros larvatus* (N = 10), *Hipposideros gentilis* (N = 14), *Hypsugo* sp. (N = 8), *Megaderma spasma* (N = 3), *Miniopterus fuliginosus* (N = 10), *Myotis horsfieldii* (N = 1), *Myotis* sp. (N = 1), *Pipistrellus abramus* (N = 20), *Rhinolophus affinis* (N = 3), *Rhinolophus macrotis* (N = 1), *Rhinolophus marshalli* (N = 4), *Rhinolophus microglossus* (N = 9), *Rhinolophus pearsonii* (N = 3), *Rhinolophus pusillus* (N = 7), *Rhinolophus siamensis* (N = 1), *Rhinolophus thomasi* (N = 9) and *Taphozous* sp. (N = 48). Four *Rousettus* sp. frugivorous bat rectal swabs were also collected.

The pan-coronavirus PCR screening identified 21 coronaviruses, among which *Sarbecovius*-related sequences were identified in seven of the eight *Rhinolophus thomasi* and in one of the *Rhinolophus siamensis* individuals. No sarbecovirus was identified in the other insectivorous or frugivorous bat species, suggesting a strong association of Vietnamese sarbecoviruses with their *Rhinolophus* sp. insectivorous hosts. One sarbecovirus originated from Khẩy lầu hamlet (Sốp cộp district, 220 km west from Hanoï), while the others were identified 80 km away, in two caves in Pha nhê hamlet (Mộc Châu district, 140 km west from Hanoï). Interestingly, two *Rhinolophus* species (i.e., *Rh. thomasi* and *Rh. siamensis*) were sampled in the same cave at the same time, highlighting the fact that different bat species live in sympatry in the same cave (Table 1). We tested oral swabs and urine samples collected from the same bat individuals for the presence of sarbecoviruses, and only two (RtVN21-193 and RtVN21-194) of the eight rectal swab-positive individuals also tested positive with oral swabs. No urine sample was positive, which suggests enteric shedding of these bat sarbecoviruses. 

### 3.2. Genomic Characterization of Rhinolophus *sp.* Sarbecoviruses

The complete genomes of seven of the eight *Sarbecovirus* rectal swab strains were obtained after *Betacoronavirus*-enriched NGS. We were not able to obtain the complete genome of the oral swab-positive sarbecoviruses.

We first compared the genetic relationships at the gene level between these strains and identified different patterns of sequence clustering depending on the gene considered. For example, the clade that comprises RtVN21-197/-200 and -201 clustered with RtVN21-29 in ORF1a and with RtVN21-192 in ORF1b and in the spike, while they clustered with a subclade formed by RtVN21-29 and -192 in the nucleoprotein (Figure 2A). These changes in tree positioning suggest that recombinations occurred during RtVN21-29 and -192 evolution, while RtVN21-197/-200/-201 and RtVN21-193/RsVN21-195 always clustered together, suggesting higher stability of their genomes. To confirm that RtVN21-29 and RtVN21-192 were subject to multiple recombination events, we performed a full exploratory recombination analysis of the seven strains of Vietnamese sarbecoviruses, and observed that only RtVN21-29 and -192 presented recombinations (Figure 2B): RtVN21-29 presented three insertions for which the donor sequences were close to RtVN21-193 and RsVN21-195 (one in the ORF1ab and two in the S1 domain of the spike), and RtVN21-192 presented four insertions originating from strains close to RtVN21-197 and -201.

Unsurprisingly, the distance matrices performed at both the gene and at the protein levels revealed that the less conserved proteins are the spike, the ORF8, the ORF3a and the ORF6 proteins, which exhibited 78%, 74%, 81% and 89% nucleotide identity between distant Vietnamese isolates, respectively (Appendix A).

Interestingly, different sarbecoviruses that belong to different clades were sampled in the same cave at the same time from two different *Rhinolophus* species (*thomasi* and *siamensis*), suggesting that at least two different lineages (one apparently restricted to *Rh. thomasi* and one shared by *Rh. thomasi* and *Rh. siamensis*) of sarbecoviruses circulate concomitantly in the area. Conversely, viruses (RtVN21-29 and -192) that shared a common ancestor (at least for the nucleoprotein) are found in bats 80 km distant from each other despite the fact that insectivorous bats are not prone to long migrations, which suggests that bats living in caves not sampled in this study should harbor other close viral species (Figure 2).

At a more global scale, Vietnamese sarbecoviruses are clustered together but are quite distant from the SARS-CoV clade, while three distinct Vietnamese clades are clearly observed, both at the whole genome and at the spike gene level (Figure 3), which is consistent with previous observations (Figure 2). Similar conclusions could be drawn when looking at the RNA-dependent RNA polymerase (RdRP), at the nucleoprotein or at the N-terminal domain (NTD) and receptor-binding domains (RBDs) of the spike (Appendix A). Interestingly, African and European *Sarbecovirus* isolates clustered together and are distinct from Asian isolates, in addition to several bat species-specific clusters that are observed for *Rh. stheno*, *Rh. sinicus* and *Rh. hipposideros*, for example. This suggests that several sarbecoviruses may be specific to a given bat species, and that the geographical specificity observed between African/European and Asian *Sarbecovirus* isolates may only be explained by the presence of different bat populations in these continents.

Phylogenetic reconstructions of the evolution history of Vietnamese sarbecoviruses, along with representative SARS-CoV- and SARS-CoV-2-related animal viruses, placed Vietnamese strains differently when looking at the RdRP or at the spike genes, suggesting that recombinations may have occurred during Vietnamese *Sarbecovirus* evolution (Figure 4). For example, RtVN21-197/-200/-201 clustered together in the RdRP, with RtVN21-29 at a more basal position, but in the same clade. In addition, RtVN21-193/RsVN21-195 again clustered together but are placed in a sister clade of the one formed by RtVN21-197/-200/-201/-192/-29. Conversely, these clusters are clearly distinct and distant from each other in the spike gene.

To evidence that recombinations have occurred during Vietnamese *Sarbecovirus* evolution, which could explain these different tree positionings, we performed a full exploratory recombination analysis, and identified 24 recombination events (Appendix A). Some events were shared by two or more Vietnamese sarbecoviruses while other were unique. For example, the recombinant fragment #343 (starting at position 22,460, which corresponds to the S1 domain of the spike) was detected in four *Rh. thomasi* sequences (RtVN21-192, -197, -200 and -201), while fragment #758 (starting at position 3318, which corresponds to the ORF1a) was restricted to RtVN21-29. Donor sequences were sometimes unique (for example, RtVN21-192 for event #31 impacting RtVN21-197, -200 and -201) or multiple, meaning that the donor sequence was not clearly identified, possibly due to the lack of an exhaustive inventory of sarbecoviruses circulating in the area.

Interestingly, two Chinese *Sarbecovirus* strains, Rhinolophus sinicus Rs4231 and Rhinolophus affinis GD2017F, identified respectively in 2013 and 2017, clustered with Vietnamese strains at the RdRP level, suggesting that similar sarbecoviruses are shared by different rhinolophid bat species living in remote areas (Appendix A). While identified in the same *Rh. thomasi* bat colony, when looking at the spike gene, the RtVN21-197/-200/-201/-192 clade was closely related to several *Rh. sinicus*, *Rh. affinis* and *Aselliscus stoliczkanus* Chinese strains while the RtVN21-193/RsVN21-195 clade was more closely related to *Rh. pusillus* and *Rh. ferrumequinum* strains from China (Appendix A). More globally, even if local clusters of bat species-specific sarbecoviruses are observed, it seems that no strict host specificity was observed during *Sarbecovirus* evolution, as illustrated by many *Rh. sinicus*-associated clusters that frequently include sarbecoviruses from other bat species. This suggests that exchanges of related sarbecoviruses are frequent among different bat species that live in sympatry in the same colonies.

### 3.3. Prediction of Zoonotic Potential of Vietnamese Sarbecoviruses

#### 3.3.1. In Silico Evaluation of RBD-ACE2 Binding Free Energy

To study the potential risk of infection with Vietnamese sarbecoviruses in different vertebrate hosts, we used an in silico pipeline to estimate the binding free energy of the RBDs of representative sarbecoviruses (including SARS-CoV and SARS-CoV-2) and the ACE2 receptors of various animal species. Representative sarbecoviruses were selected from each of the three major clades in the RBD phylogenetic tree (Appendix A). Animal species were selected to cover the diversity of rhinolophid bats that carry sarbecoviruses, and various domestic (pigs, cattle, dogs, cats, horses, chickens) and wild (rodents, pangolins, civets, porcupines, mustelids) animals that have been demonstrated or suspected to act as intermediate hosts for certain sarbecoviruses were selected.

Although SARS-CoV and SARS-CoV-related bat and civet sarbecoviruses showed predicted good affinities for human, pig, cat, raccoon dog, horse, porcupine, mustelid, rabbit and alpaca ACE2s, more distant SARS-CoV-related bat sarbecoviruses (such as *Rh. malayanus* BANAL-20-116 or Vietnamese *Rh. thomasi* and *Rh. siamensis* viruses) showed weak to very weak affinity for any human or animal ACE2 tested (Figure 5). Interestingly, we identified two large deletions in the receptor-binding motif of the RBDs of Vietnamese sarbecoviruses involving known amino-acid contact residues between virus RBDs and ACE2 receptors (Appendix A), which are likely responsible for the apparent inability of our viruses to bind efficiently to any animal ACE2.

#### 3.3.2. ACE2-Dependent Entry Assay of Vietnamese Sarbecoviruses

To confirm these in silico predictions, we generated lentiviral particles pseudotyped either with the spike protein of one representative strain of Vietnamese sarbecoviruses (i.e., RtVN21-192, 193, -201 or -29) or with SARS-CoV-2 Wuhan (strain BetaCoV/France/IDF0372/2020, GISAID accession number EPI_ISL_406596). We tested the ability of these pseudotypes to enter into 293T-hACE2 cells (Figure 6) and observed that none of the Vietnamese strains were able to enter cells, unlike SARS-CoV-2. This result confirmed the in silico predictions and suggests that the deletion observed in the receptor-binding motif (Appendix A) influences the ability of Vietnamese sarbecoviruses to use the human ACE2 receptor to infect humans.

## 4. Discussion

Southeast Asia is a major hotspot for coronavirus emergence due to the huge richness in bat species that live and interact with high densities of human and domestic animal populations. This proximity is a key component of the processes of emergence of bat-borne zoonotic viruses. Direct (bat-to-human) or indirect (via an intermediate host, also known as a bridge host) spillovers could occur, and may lead to epidemics or pandemics, as observed in recent decades. Monitoring the circulation of coronaviruses in their bat reservoirs before their emergence in human or in domestic animal populations and understanding the molecular and epidemiological factors that shape spillover events remain challenges for public health.

In a work aiming at modeling the risk of *Sarbecovirus* spillover to humans, Sánchez et al. identified Northern Vietnam as one of the areas of Southeast Asia that harbors the highest diversity of bat-associated SARS-related coronaviruses [71]. Similarly, Hassanin et al. predicted that bats from Northern Vietnam could harbor SARS-CoV-like viruses [72]. Apart from these predictions, sarbecoviruses were never identified from Vietnam before this study, although extensive sampling efforts have been made. Here, we report the first identification and characterization of *Rhinolophus* sp. sarbecoviruses in Northern Vietnam. Numerous insectivorous bat species were tested for the presence of coronaviruses, including horseshoe bats, which are known to be a major reservoir of sarbecoviruses [7], and only *Rhinolophus* bats (i.e., *Rh. thomasi* and *Rh. siamensis*) were found to be positive for sarbecoviruses. This finding revealed a strong association of Vietnamese sarbecoviruses with their *Rhinolophus* hosts, and indeed it has been proposed that such strong association reflects the long coevolution of bat-associated sarbecoviruses with their bat reservoirs [7].

Phylogenetic characterization of Vietnamese sarbecoviruses revealed three major findings. First, the changes in tree positioning and the recombination events observed between the different strains suggest that recombination was a major driver of *Sarbecovirus* evolution and a possible source of novel viruses, as supposed for SARS-CoV-2 [73]. Second, the observation that different *Rh. thomasi* sarbecoviruses (RtVN21-193 vs. RtVN21-197, -200, -201) belonging to different clades were sampled in the same cave at the same time shows that different lineages of viruses are circulating concomitantly in the same area, which favors recombination events. Finally, the fact that the same virus (presenting among others the same receptor-binding domain) is found in different *Rhinolophus* species (RtVN21-193 from *Rh. thomasi* and RsVN21-195 from *Rh. siamensis*) suggests that cross-species transmission among bats is a major trait of sarbecoviruses (Figure 4, [25,72]). This observation suggests that, even if sarbecoviruses coevolved with their bat hosts, transmission of viruses across different bat species is frequent, which favors the emergence of novel viruses through recombinations or other evolutionary processes. This in part likely reflects bat roosting behavior and propensity to share the same or close habitats, which again favor the exchange of viruses and recombination. Understanding the drivers of the emergence of new pathogens is one of the major challenges in the ecology of infectious diseases. It has been proposed that Southeast Asia (comprising, among others, Southern China, Northern Laos and Northern Vietnam) constitutes a hotspot for coronavirus emergence because of a favorable bioclimatic environment and a high diversity of bat species [61,72]. The presence of this hotspot is due in part to the climate, the landscape and the human population density of the region [74,75]. In a recent study, Beyer et al. reported, for example, the impact of climate change on the geographical distribution of bats, which has resulted in an increase in bat species richness in specific locations that could have been the origin of *Sarbecovirus* emergence [76].

In line with previous observations that showed that coronavirus carriage in bats was mainly enteric [27,46], we observed that few (25%) positive individuals presented a positive oral swab, and none presented a positive urine sample. Indeed, a recent study showed that the ACE2 mRNA expression level within bats greatly varies between organs and is higher in the intestine than in the kidney, for example [77]. Enteric shedding of bat sarbecoviruses therefore constitutes a risk of spillover, especially when considering people frequently exposed to bat guano such as guano collectors, people working in guano farms or people visiting caves who are exposed to guano dust [47,49]. Knowing the ability of Vietnamese *Rhinolophus* sp. sarbecoviruses to infect humans is therefore crucial. There is currently no evidence to suggest that these viruses pose a significant threat to public health. Therefore, to decipher the zoonotic potential of these sarbecoviruses, we evaluated their ability to enter mammalian cells using ACE2 receptors.

First, we evaluated in silico the interaction strength between the RBDs of Vietnamese sarbecoviruses and different mammalian ACE2 receptors. We did not restrict the analysis to the bat/human interface because, even if the intermediate host for SARS-CoV-2 emergence is yet unknown, past emergence of coronaviruses frequently implies the presence of a bridge host [6], such as domestic (dogs, pigs, dromedary camels) or wild animals (palm civets, raccoon dogs) that are bred as food resources for humans in several Southeast Asian countries, including Vietnam. The analysis indicated no robust interaction between Vietnamese *Sarbecovirus* RBDs and human or wild or domestic animal ACE2. The predictive value of these results is supported by the accurate prediction of the positive interaction of human ACE2 with SARS-CoV-2 RBDs, along with the strong affinity of SARS-CoV for swine ACE2, as previously reported [78,79]. When looking at the receptor-binding motif (RBM) of Vietnamese sarbecoviruses, we observed two important deletions that may directly impact the affinity of their RBDs to mammalian ACE2 receptors by changing the structure of the spike protein. In nature, hundreds of sarbecoviruses like those described here have been found in bats, but only a small fraction of them have the ability to infect cells using ACE2. It has been proposed that bat sarbecoviruses be classified according to their RBD sequence and its subsequent ACE2 utilization [42]. Guo et al. reported that among the four clades of sarbecoviruses that can be identified, only Clade I viruses (harboring no deletion in the RBM) can utilize endogenous ACE2 for cell entry, and only these viruses are able to replicate in vitro or in vivo. Our viruses fall within Clade II of this classification, and initial attempts to isolate such viruses from field samples or propagate in vitro from reverse genetics were unsuccessful. Authors have also showed that in the presence of high levels of trypsin, some of the spike proteins of Clade II viruses were capable of mediating infection and replication in human (including Caco-2 intestine cell line) and bat intestine primary cells while TMPRSS2 and furin endogenous proteases had no effect on Clade II virus entry [42].

Apart from the rich trypsin environment, other factors such as the temperature may impact the infection and replication abilities of sarbecoviruses. Partridge et al. observed that the whole spike (and not the S1 or RBD domains) of SARS-CoV-2 exhibits a temperature-dependent cellular interaction; however, the underlying mechanisms are not fully understood [80]. Bats are the only mammalian species that is able to fly, which required evolutionary adaptations for them to survive the elevated body temperatures occurring during flight [81,82]. Therefore, it can be proposed that the unique temperature regulation of bats may impact the ability of bat sarbecoviruses to bind efficiently to bat cells, thus preventing other homeothermic species from being infected by bat sarbecoviruses that coevolved over long periods with their hosts. In line with the presence of the two deletions in the receptor-binding motif, such findings suggest that these viruses are likely unable to cross the species barrier and to pose a risk to human or animal health.

In addition to the presence of two deletions in the RBM, several amino-acid substitutions were observed at the contact residues. The plasticity of the spike protein of sarbecoviruses, which allows them to adapt to a broad range of ACE2 receptors, is now well known, and such substitutions at contact residues allow several viruses to increase their affinity for ACE2 [27]. Molecular evolution analyses indicated that these key residues are under positive selection, suggesting that the spike protein of bat sarbecoviruses and mammalian ACE2 may have coevolved over a long time and adapted to selection pressure from each other [83,84]. The substitutions observed in Vietnamese contact residues may therefore be a signature of the inability of a given sarbecovirus to interact efficiently with ACE2, as observed, for example, for Rhinolophus stheno RsYN09 [85], Rhinolophus affinis YN2020A [55], Rhinolophus sinicus Rs4247, Rhinolophus ferrumequinum Rf4092 [26] and Rhinolophus pusillus F46 [86] sarbecoviruses. These simple features may serve as preliminary markers of the ability or inability of a new sarbecovirus to bind to mammalian host cells via ACE2, and possibly infect them, as studied in depth when SARS-CoV-2 emerged [87].

Since the RBD alone cannot predict the behavior of the whole spike in the context of a mature virion, especially given the presence of the two deletions in the receptor-binding motif, we constructed pseudotyped lentiviruses to measure the interaction of the spike protein of Vietnamese coronaviruses with human ACE2. The results of entry assays confirmed that Vietnamese sarbecoviruses were not able to efficiently interact with hACE2, preventing their entry into 293T human cells. Indeed, hundreds of sarbecoviruses have been found in bats, but only a fraction of them have the ability to use ACE2 [42]. It can be hypothesized that other proteins act as alternative receptors, in complement to or in place of ACE2 [88]. For example, an ACE2-independent entry of SARS-CoV-2 into different human cells via the neuropilin -1 [89] or glycosaminoglycans [90], which are much more expressed at the surface of certain cells (including intestine and kidney cells) than hACE2, has been reported [91].

While apparently lacking the ability to interact with human (and animal) ACE2, and hence unlikely to be zoonotic without the acquisition of several sequence changes, Vietnamese sarbecoviruses enrich the pool of viruses circulating in the area and present opportunities for homologous recombination that could lead, for example, to a new virus able to cross the species barrier. Also, recombination could lead to increased virulence compared with the parental strains [92], to the capability to escape neutralization by antibodies able to neutralize the parental strain [93] or to changes in host or tissue tropism [94]. Therefore, identifying novel coronaviruses should not be restricted to the (bat) reservoir but also should include any potential bridge hosts that could be infected by emerging coronaviruses. The identification of such hosts could easily be implemented by applying a straightforward strategy that includes in silico binding analyses (conducted over large RBD/ACE2 datasets) in order to capture a global picture of the host range and risk of cross-species transmission of emerging sarbecoviruses, and to improve the surveillance of specific interfaces at higher risk of emergence.

## 5. Conclusions

Southeast Asia hosts a great diversity of wildlife and is undergoing important climate and land-use changes that can increase contact between wildlife and humans. Continued and expanded surveillance of bat populations and other key wild and domestic animals in Southeast Asia, and especially Vietnam, is a crucial component of future pandemic preparedness and prevention. Such programs should include the surveillance of sentinel interfaces and the use of novel and easy-to-implement molecular and bioinformatics tools to identify high-risk wildlife-to-human interfaces.

## Figures and Tables

**Figure 1 viruses-15-01897-f001:**
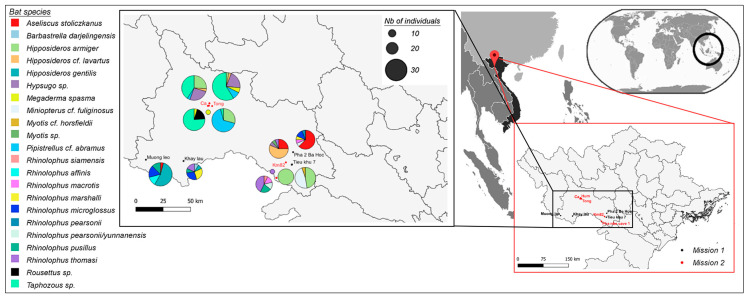
Bat sampling in Northern Vietnam, 2021. Maps were constructed with Quantum GIS software (https://qgis.org/en/site/ accessed on 6 September 2023). Sampling performed during February–March 2021 is presented by a black dot (Mission 1), and sampling performed during July 2021 is presented by a red dot (Mission 2). The size of the pie charts is proportional to the number of bats sampled in each location. Bats were captured using mist nets and hand nets. Each captured bat was weighted using a Pesola spring scale, then the bats were individually kept in porous humidified cotton bags and placed in a cool dry place for sampling within six hours. The following basic set of samples was collected from each bat, and included saliva (oropharyngeal swab), feces (fresh fecal sample) or rectal swab, blood (serum; rbc/wbc pellet) and urine (free catch method or urogenital swab). After sampling, bats were released at the collection site.

**Figure 2 viruses-15-01897-f002:**
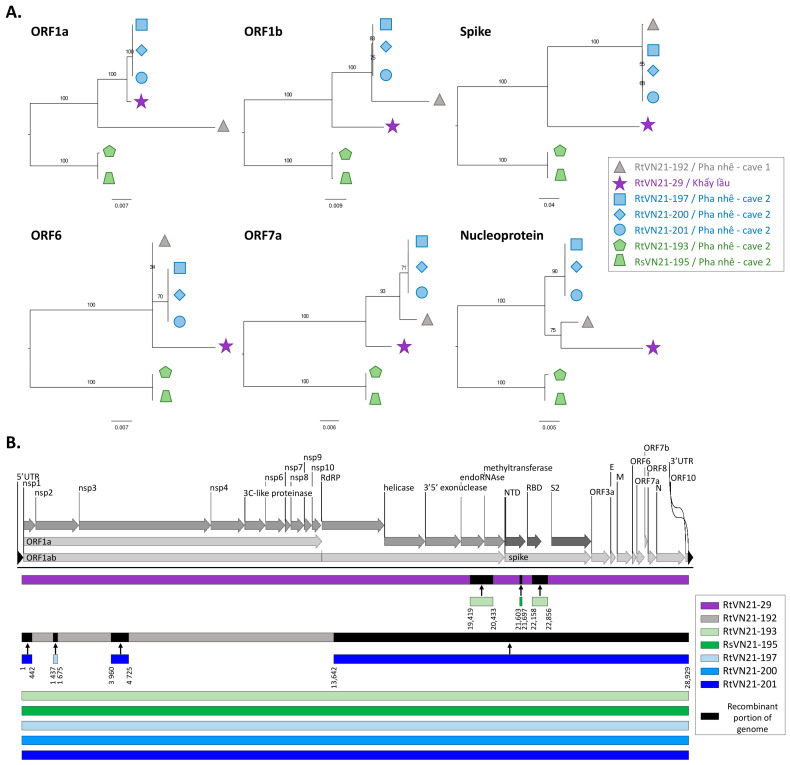
Genomic characterization of Vietnamese sarbecoviruses. (**A**) Phylogenetic analysis of ORF1a, ORF1b, spike, ORF6, ORF7a and nucleoprotein genes of Vietnamese sarbecoviruses. Strains are represented by symbols and colored according to their position in the different clades. (**B**) Recombination analysis of Vietnamese sarbecoviruses. Recombinant fragments and corresponding breakpoints are indicated below each backbone and colored according to the donor genome sequence. Breakpoint coordinates correspond to the position in the alignment available at 10.6084/m9.figshare.23821428 (accessed on 6 September 2023). SARS-CoV-2 genome (NC_045512) served as reference for protein domains.

**Figure 3 viruses-15-01897-f003:**
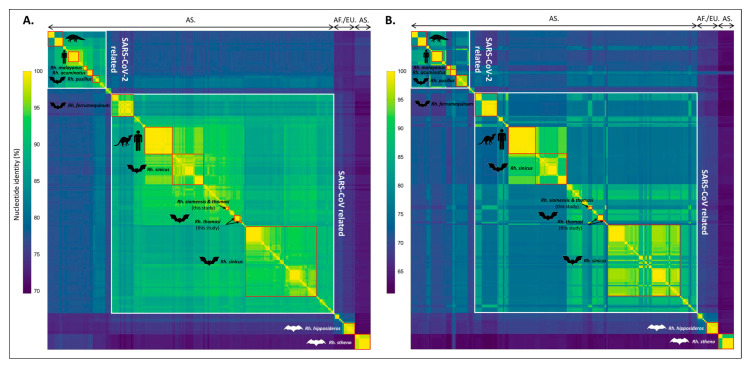
Heatmap of nucleotide identities observed at the whole genome (**A**) and at the spike (**B**) levels between Vietnamese and 221 representative *Sarbecovirus* genomes. Aligned sequences were sorted by similarity. Low-to-high nucleotide identities are represented from blue to yellow. SARS-CoV- and SARS-CoV-2-related sequences are highlighted by white squares, and bat-specific clusters are highlighted by red squares. Identity matrices were calculated with CLC Main Workbench 21.0.4 (Qiagen) and the heatmap was created using GraphPad Prism 9.5.1 (GraphPad software). AS = Asian isolates; AF/EU = African/European isolates; Rh. = *Rhinolophus*. The complete identity matrices are presented in Appendix A.

**Figure 4 viruses-15-01897-f004:**
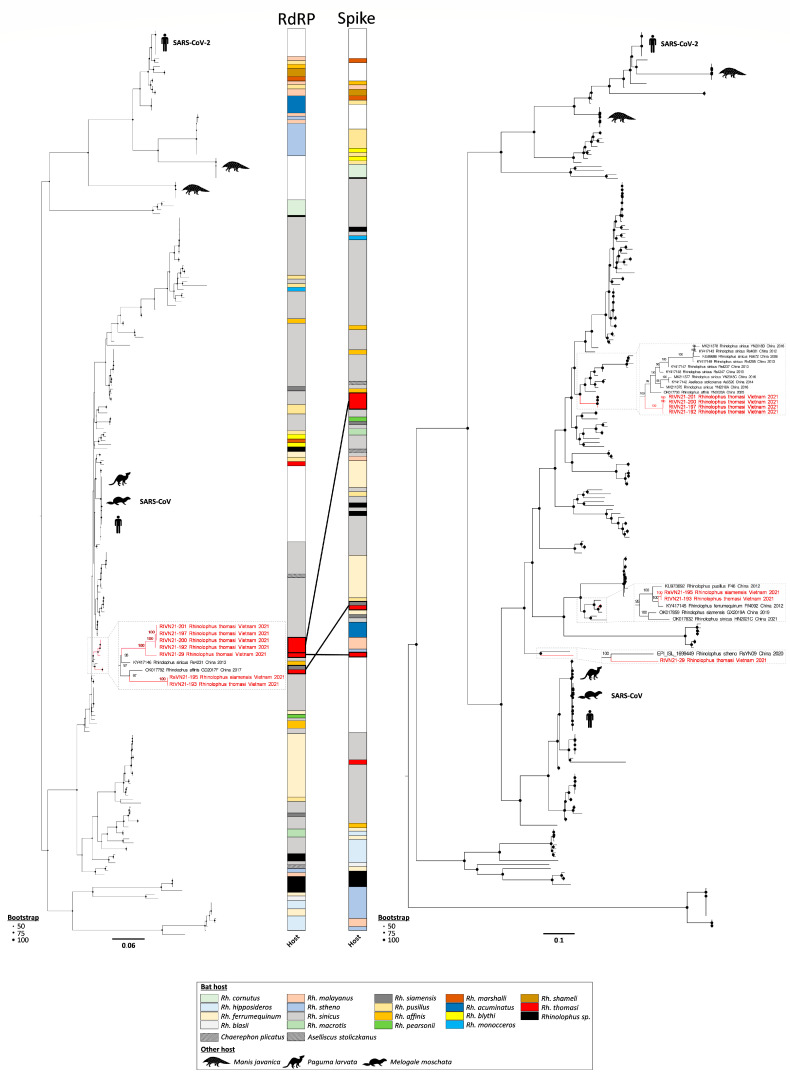
Phylogenetic analysis of Vietnamese and 221 representative sarbecoviruses. Trees were inferred for the RdRP and spike genes, and midpoint-rooted for clarity. Bootstraps above 50 are represented by black dots. Sarbecoviruses from *Rhinolophus* sp. bat species are represented by colored squares while *Chaerephon plicatus* and *Aselliscus stoliczkanus* sarbecoviruses are represented by dashed lines. Complete phylogenetic reconstructions are available in Appendix A.

**Figure 5 viruses-15-01897-f005:**
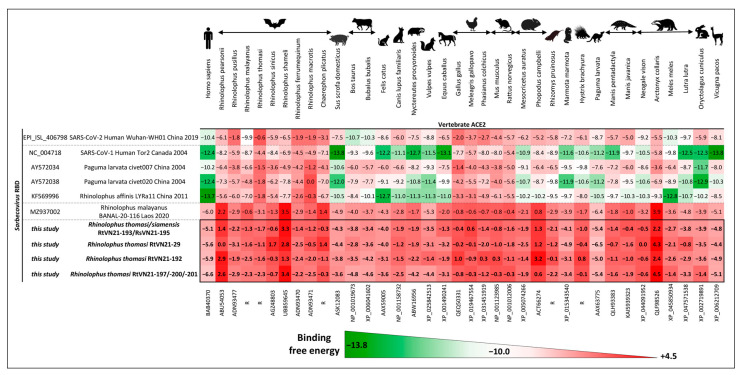
Estimation of the binding free energy of representative sarbecovirus RBDs for different animal ACE2 receptors. Scale was set up based on the binding free energy of human SARS-CoV-2 with human ACE2 (−10.0; white). Strong predicted binding free energy (below −10.0) is represented by shades of green while weak binding free energy (above −10.0) is represented by shades of red. “R” represents ACE2 sequence that was manually reconstructed from public and private datasets (see Section 2).

**Figure 6 viruses-15-01897-f006:**
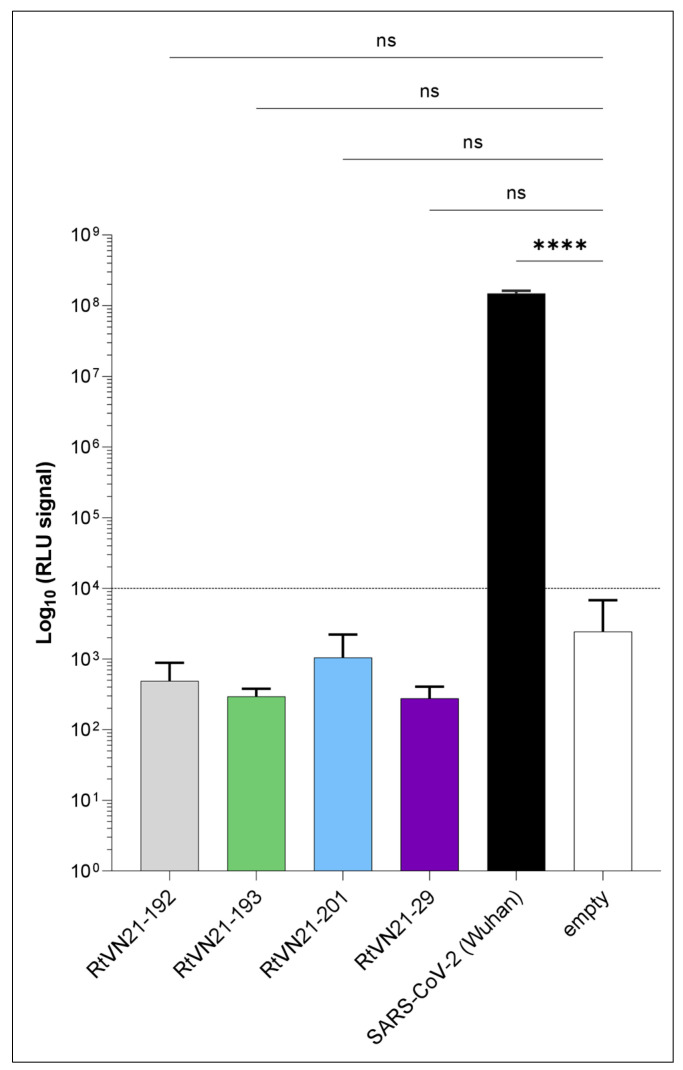
ACE2-dependent entry assay of *Rhinolophus* sp. sarbecovirus pseudotypes. Results of a single experiment performed in triplicate are expressed in relative luminescence units (RLUs) produced by the firefly luciferase expressed from the lentiviral vector. Error bars indicate SD. Statistical significance was assessed using one-way ANOVA and was compared with the RLU signal of an empty vector as a reference: ns: non-significant; ****: *p* < 0.0001.

**Table 1 viruses-15-01897-t001:** Characteristics of *Sarbecovirus*-positive rectal swabs.

Virus Strain	Province	District	Commune	Hamlet	Latitude (DD.ddddd’)	Longitude (DD.ddddd’)	Elevation (meters)	Bat Species	Bat COI Accession No.	Bat Weight (g)	Genome Coverage	CoV Accession No.
RtVN21-29	Sơn La	Sốp cộp	Mường và	Khẩy lầu	20.882820	103.695280	936	*Rhinolophus thomasi*	OR427971	10.7	100%	OR261262
RtVN21-192	Sơn La	Mộc Châu	Lóng Sập	Pha nhê/cave 1	20.736283	104.504630	1059	*Rhinolophus thomasi*	OR435217	8.0	100%	OR261263
RtVN21-193	Sơn La	Mộc Châu	Lóng Sập	Pha nhê/cave 2	20.734502	104.505343	1073	*Rhinolophus thomasi*	OR427972	9.3	100%	OR261264
RtVN21-194	Sơn La	Mộc Châu	Lóng Sập	Pha nhê/cave 2	20.734502	104.505343	1073	*Rhinolophus thomasi*	OR427973	8.2	10.35%	nd
RsVN21-195	Sơn La	Mộc Châu	Lóng Sập	Pha nhê/cave 2	20.734502	104.505343	1073	*Rhinolophus* *siamensis*	OR427974	7.1	100%	OR261265
RtVN21-197	Sơn La	Mộc Châu	Lóng Sập	Pha nhê/cave 2	20.734502	104.505343	1073	*Rhinolophus thomasi*	OR427975	7.9	100%	OR261266
RtVN21-200	Sơn La	Mộc Châu	Lóng Sập	Pha nhê/cave 2	20.734502	104.505343	1073	*Rhinolophus thomasi*	OR427976	8.0	100%	OR261267
RtVN21-201	Sơn La	Mộc Châu	Lóng Sập	Pha nhê/cave 2	20.734502	104.505343	1073	*Rhinolophus thomasi*	OR427977	7.3	100%	OR261268

## Data Availability

The complete genome sequences of Vietnamese sarbecoviruses and bat-associated COIs were deposited in GenBank (accession numbers OR261262-68 and OR427971-77, respectively). GISAID identifier: EPI_SET_230623sh (DOI: 10.55876/gis8.230623sh). All genome sequences and associated metadata in this dataset are published in GISAID’s EpiCoV database. The alignments used to reconstruct phylogenies presented in this study and the complete RDP5 recombination analysis are openly available in FigShare (DOI: 10.6084/m9.figshare.23821428 and 10.6084/m9.figshare.23821719, respectively).

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
