# Peer review of "Genotype and Phenotype Characterization of Rhinolophus sp. Sarbecoviruses from Vietnam: Implications for Coronavirus Emergence"

_viruses, 2023, doi:10.3390/v15091897_

Round 1
Reviewer 1 Report
In this article, the authors have reported seven sarbecoviruses found in Rhinolophus thomasi bats living in northern Vietnam. The authors concluded that ‘Recombination and cross-species transmission between bats seemed to constitute major drivers of virus evolution’ and that ‘these viruses are likely restricted to their bat host’. Although the results are interesting, most figures are difficult to read or interpret, thus casting doubt on the quality of the genomes and analyses generated for this study. To avoid misinterpretation, I recommend several changes in the table and figures (see below) as well as additional analyses to evidence recombination.
Title
What does ‘sarbecovirus phenotype’ mean? A few explanations should be developed in Introduction.
Introduction
- The bibliography is good but it could be slightly improved. In PubMed, I found 17 articles with the words bat + coronavirus + Vietnam. Among them, I consider that the two following articles should be cited in Introduction and discussed in the manuscript:
1. Identification and characterization of Coronaviridae genomes from Vietnamese bats and rats based on conserved protein domains. Phan MVT, Ngo Tri T, Hong Anh P, Baker S, Kellam P, Cotten M. Virus Evol. 2018 Dec 15;4(2):vey035. doi: 10.1093/ve/vey035.
To my knowledge, this is the first genome study on Coronaviridae from Vietnamese bats. Even if it deals with South Vietnam, it should be mentioned in Introduction.
2. Inferring the ecological niche of bat viruses closely related to SARS-CoV-2 using phylogeographic analyses of Rhinolophus species. Hassanin A, Tu VT, Curaudeau M, Csorba G. Sci Rep. 2021 Jul 12;11(1):14276. doi: 10.1038/s41598-021-93738-z.
In this study, the authors have predicted SARS-CoV-like viruses in northern Vietnam. In the Abstract, they also wrote that ‘In the climatic transitional zone between the two ecological niches (southern Yunnan, northern Laos, northern Vietnam), genomic recombination between highly divergent viruses is more likely to occur.’ These two points should be more discussed in the manuscript.
Materials and Methods
- A map showing sampled locations in Vietnam should be provided by the authors, including all negative studied locations.
- L109. For ethical reasons, the authors should indicate whether bats were euthanized or released into the wild during this study.
- The authors should explain how bats were identified at the species level. Indeed, it seems that Rhinolophus thomasi is problematic in northern Vietnam (Mao et al 2019; see reference below). To confirm species identification, I therefore recommend to extract from Illumina reads the cytochrome c oxidase subunit 1 gene for all the seven bats found positive for coronavirus. Accession numbers of these new sequences should be included in table 1.
Reference: Resolving evolutionary relationships among six closely related taxa of the horseshoe bats (Rhinolophus) with targeted resequencing data. Mao X, Tsagkogeorga G, Thong VD, Rossiter SJ. Mol Phylogenet Evol. 2019 Oct;139:106551. doi: 10.1016/j.ympev.2019.106551.
- Alignments used for phylogenetic analysis and genome/gene trees should be deposited in dryad.com
- ‘IQTree’ should be replaced by ‘W-IQ-TREE’ throughout the main text.
- The origins of ACE2 receptor sequences (accession numbers) used for in silico predictions should be provided as supplementary table.
Results and Discussion
- Figure 1 shows six trees reconstructed (maximum likelihood?) using the seven Vietnamese sarbecoviruses and different genes (ORF1a, ORF1b, spike, ORF6, ORF7a and nucleoprotein).
According to the authors (L223-L227), ‘these changes in tree positioning suggest that recombinations have occurred during RtVN21-29 and -192 evolution, while RtVN21-197 / -200 / -201 and RtVN21-193 / -195 always clustered together, suggesting higher stability of their genomes.’ My point of view is that only a different placement of RtVN21-192 is supported between ORF1a tree and other trees. Other relationships are poorly supported. This means that several recombination events cannot be proposed based on tree discordance. Another problem is that RtVN21-192 shows a very long branch in the ORF1a tree suggesting that its different placement in the ORF1a tree could be explained by something other than recombination, including long branch attraction (LBA) artefacts or sequencing errors.
According to Mat & Meth, the IQ-TREE web server was used for phylogenetic analysis. IQ-TREE is a fast maximum likelihood method very useful for large data sets, but generally less reliable on very small data sets, as the one used by the authors. To limit the impact of LBA artefacts, I strongly recommend to add published sarbecovirus sequences in the phylogenetic analyses of Figure 1.
In regard to the hypothesis involving possible sequencing errors, the authors wrote that ‘conventional PCR and Sanger sequencing were used to fill the gaps in the genomes’ (L137). To guarantee the quality of the genomes sequenced in this study, I would like to see a table describing for each of the seven coronavirus genomes, the number of Illumina reads available and the number of reads mapped in the coronavirus genome assembly. The authors should also indicate the number of gaps obtained after genome assembly based on Illumina reads. The primer sequences used to fill the gaps by PCR should be provided in a supplementary table.
- Figure 2 shows that Vietnamese coronaviruses are included in two clusters, whereas the tree of Supp. Figure 5 shows three clusters (as mentioned in the Abstract). The four red clusters indicated in the Spike tree of Figure 3 are assumed to be a mistake (but it needs to be checked). These inconsistent results need clarification in the main text.
- The quality of Figure 3 is very bad and its interpretation is impossible because virus names were not reported after terminal branches. The colored bars suggest very discordant relationships, but the results were not discussed in the main text. I would prefer to see a reduced and modified version of the tree shown in Supp_Figure 5. I strongly recommend to remove most of the 221 genomes used in the alignment. Indeed, it seems obvious that many of these sequences are identical or very similar, including many SARS-CoV-2, SARS-CoV, pangolin coronaviruses, etc. Using a simple distance criterion, the alignment could be reduced to about 100 high-quality genomes for phylogenetic analysis, making interpretation easier.
- Figure 4 shows very weak affinity between the spike RBD of Vietnamese bat sarbecoviruses and most mammalian ACE2 receptors. These results suggest that Vietnamese bat sarbecoviruses did not use ACE2 receptor for cell entry. Does this mean that Vietnamese bat sarbecoviruses use a different receptor than ACE2? This point should be more discussed.
- The authors have proposed several times ‘that recombinations may have occurred during Vietnamese Sarbecovirus evolution (Figure 3)’ (L268). However, they did not describe the viruses and genomic fragments involved in recombination. To evidence recombination, I recommend to use a more specific method, such as 3SEQ, Simplot, etc.
- L284-285: ‘This suggests that exchanges of related sarbecoviruses are frequent among different bat species that live in sympatry in the same colonies.’ Paradoxically, all the viruses in this study have been described in a single bat species. Do the authors think that many other bat species are likely to harbor coronaviruses in Vietnam? This question needs to be discussed.
Reviewer 2 Report
This is a well-written study of 7 sarbecoviruses sequenced from Thomas's horseshoe bats in Northern Vietnam. The SARS-CoV-2 pandemic highlighted how little is known about the genetic diversity and evolution of sarbecoviruses in zoonotic reservoir hosts, especially bats. Surveys such as this are needed to understand patterns of recombination, genetic motifs, and spillover.
Line 154: Which substitution model did you use?
Page 5/Table 1: Were the Sarbecovirus rectal swabs from Phanhe cave 2 all collected on the same date? If so, this could represent a stochastic sampling of a transient outbreak and not necessarily broadly higher infection rates in the Thomas species.
Recombination: it would be helpful to have a SimPlot to visualize recombination patterns
One of the interesting findings is that there were genetically distinct sarbecoviruses co-circulating in cave 2. But just to be clear, you did not observe any recombinants of these two parental strains, correct?
Line 250-254: "Interestingly, African and European Sarbecovirus isolates clustered together and are dis- 250 tinct from Asian isolates, in addition to several bat species-specific clusters that are ob- 251 served for Rh. stheno, Rh. sinicus or Rh. hipposideros for example. This suggests that several 252 sarbecoviruses may be specific for a given bat species, and that the geographical specificity 253 observed between African/European and Asian Sarbecovirus isolates may only be ex- 254 plained by the presence of different bat populations in these continents. " This is an interesting idea, to consider the roles of geography versus host speciation in shaping virus gene flow. Is it possible to perform some statistical tests of this hypothesis?
